# Community Health Representatives as Trusted Sources for Increasing Representation of American Indian Communities in Clinical Research

**DOI:** 10.3390/ijerph20054391

**Published:** 2023-03-01

**Authors:** Samantha Sabo, Naomi Lee, Grant Sears, Dulce J. Jiménez, Marissa Tutt, Jeffersson Santos, Omar Gomez, Nicolette Teufel-Shone, Marianne Bennet, J. T. Neva Nashio, Fernando Flores, Julie Baldwin

**Affiliations:** 1Center for Health Equity Research, Northern Arizona University, Flagstaff, AZ 86011, USA; 2Department of Chemistry and Biochemistry, Northern Arizona University, Flagstaff, AZ 86011, USA; 3River People Health Center, Scottsdale, AZ 85256, USA; 4White Mountain Apache Tribe CHR Program, Whiteriver, AZ 85941, USA; 5Colorado River Indian Tribes, Parker, AZ 85344, USA

**Keywords:** American Indian, clinical trial participation, community health representatives, cultural humility, equity

## Abstract

Indigenous and American Indian Alaskan Native (AI/AN) community members are systematically underrepresented in clinical trial research. This paper focuses on exploratory steps to partner with Native Nations of Arizona to engage Community Health Representatives (CHR) as a trusted source for building COVID-19 clinical trial research, including vaccine trials awareness. CHRs are frontline public health workers who apply a unique understanding of the experience, language, and culture of the population served. This workforce has entered the spotlight as essential to the prevention and control of COVID-19. Methods: Three Tribal CHR programs were engaged to develop and refine culturally centered educational materials and a pre-post survey using a consensus-based decision-making approach. CHRs used these materials in brief education sessions during regular client home visits and community events. Results: At 30 days post CHR intervention, participants (N = 165) demonstrated significantly increased awareness about and ability to enroll in COVID-19 treatment and vaccine trials. Participants also described a significant increase in trust in researchers, decreased perceived barriers related to cost for participation in a clinical trial, and improved belief that participation in a COVID-19 clinical trial for treatment was considered a benefit to American Indian and Alaskan Native people. Conclusion: CHRs as trusted sources of information, coupled with culturally centered education materials designed by CHRs for CHR clients, demonstrated a promising approach to improved awareness of clinical trial research generally and COVID-19 trials specifically among Indigenous and American Indian community members of Arizona.

## 1. Introduction

Historical and contemporary inclusion of racial and ethnic diversity is paramount to eliminating health disparities and achieving health equity. Indigenous and American Indian and Alaskan Native (AI/AN) people are systematically underrepresented in clinical trial research [1,2]. In the midst of the COVID-19 pandemic, lack of engagement of AI/AN communities in COVID-19 vaccine trials became apparent. Compared to the US population, AI/AN people were wholly underrepresented in the COVID-19 vaccine trials [3], yet experienced the disease at twice the rate, were four time more likely to be hospitalized and almost three time more likely to die of COVID-19 compared to their white age adjusted counterparts. 

Several social and structural barriers exist for Indigenous and AI/AN representation within clinical trial research, including western scientific constructs that are most often conducted by institutions from dominant non-Indigenous cultures [1]. These institutions, in turn, often perpetuate systemic and oppressive western values, such as individualism and competitiveness, that have come to dominate society and structures within Indigenous homelands [1]. Moreover, western institutions have historically overlooked Indigenous knowledge systems as they are fundamentally holistic, experience-based, narrative-based, and relational [4,5]. Other challenges for AI/AN clinical trial research participation include distrust of researchers, lack of Indigenous representation in the clinical research design and implementation, and clinical trial research materials and protocols that are inappropriate for unique Indigenous culture and values [6,7]. Furthermore, western institutions fail to plan for social and structural exclusions such as lack of cell phones and geographic distance from the study sites, which can contribute to participants never learning the results of research they were involved in [6].

Conducting culturally centered clinical trial research in collaboration with AI/AN communities has been evidenced as the primary facilitator for AI/AN engagement in trials [7]. Culturally centered clinical trial research aims to meet the health priorities identified by AI/AN communities through culturally appropriate study design, appropriate study materials and recruitment strategies, and leadership and involvement from AI/AN academic and community research staff [6,7]. Despite efforts to eliminate incongruent representation of AI/AN in the COVID-19 trials, enrollment disparities remain. 

By the beginning of September of 2020, the National Institutes of Minority Health and Health Disparities (NIMHD) had launched several priority initiatives to promote community-engaged research to respond to the COVID-19 pandemic in the United States. This paper describes the results of a partnership between two of these NIMHD-funded COVID-19 response initiatives. Specifically, this paper focuses on exploratory steps to leverage these initiatives to engage CHRs as a trusted source for building awareness and trust in information about COVID-19 clinical trial research, including vaccine trials. In Arizona, the CHR workforce has entered the spotlight as essential to COVID-19 prevention and control [8]. We specifically describe a highly participatory and trusted partnership with three tribally operated CHR programs to engage American Indian (AI) community members in education about clinical trial participation generally, and COVID-19 research participation specifically. Community Health Workers (CHWs), also known as CHRs, are frontline public health workers who apply a unique understanding of the experience, language, and culture of the population served. CHRs are tribally employed CHWs serving American Indian and Alaskan Native communities throughout the United States [9]. CHRs are highly trained, well-established standardized workforce, and serve the medical and social needs of AI/AN communities [8,10]. This manuscript uses terms in today’s society: Native Americans, American Indian and Alaska Native (AI/AN), Indigenous, and Native Nations. When referring the larger population, we use Indigenous and/or AI/AN. However, when referring to the study participants from Arizona, we use Native Nation, AI, or the specific community. 

## 2. Materials and Methods

The NIMHD Community Engagement Alliance (CEAL) Against COVID-19 Disparities initiative offered funding to 11 U.S. states, including Arizona, to conduct outreach and engagement efforts in ethnic and racial minoritized communities disproportionately affected by the COVID-19 pandemic [11]. The Arizona CEAL Consortium is a collaboration of the three Arizona public Universities (Northern Arizona University, University of Arizona, Arizona State University), the Mayo Clinic of Arizona, and the Arizona Community Health Worker Association [12]. In partnership with members and leaders of African American, Hispanic/Latinx, and American Indian communities, the Arizona CEAL aims to provide trustworthy information through active community engagement and outreach, with the goal of building long-lasting partnerships as well as improving diversity and inclusion in the research response to COVID-19. 

The Southwest Health Equity Research Collaborative (SHERC) is a NIMHD grant-funded cooperative agreement operated by the Center for Health Equity Research at Northern Arizona University. The goal of the SHERC is to increase basic biomedical, clinical, and behavioral research at NAU to address health disparities and advance health equity among diverse populations of the southwestern United States [12]. In September 2021, SHERC received an administrative supplement aimed at building trust and awareness to increase Arizona Native Nation participation in COVID-19 vaccine trials and vaccines. Methodological details and results related to the broader study results are reported in this special issue [13]. Together, researchers leveraged expertise and community partnerships to engage in culturally centered community-based research to strengthen vaccine and clinical trial awareness among American Indian communities of Arizona disproportionately impacted by COVID-19.

### 2.1. Step 1: Formative Research through CEAL Community Health Representative (CHR) Survey and Focus Groups

The CEAL project focused on engaging the broader Community Health Worker workforce in Arizona, inclusive of the tribally employed CHR workforce, to build workforce capacity to address COVID-19 prevention and vaccine uptake. CEAL researchers developed a CHR survey and focus group interview guide to engage CHRs (N = 45) in understanding topics important to the development of COVID-19 education materials. [12,14] Topics included contemporary client experiences with COVID-19, misinformation and myths, concerns and benefits of the COVID-19 vaccine, CHR core roles during COVID-19, and CHR COVID-19 training needs.

Although not the focus of this paper, CEAL focus group data were audio recorded and transcribed verbatim by Indigenous and non-Indigenous research staff. Using a code book, research staff independently coded focus group transcripts and, through a process of consensus, identified common themes for each focus group discussion. Before participating in the focus groups, CHRs also completed a brief survey that collected demographic information, vaccine status, and COVID-19 prevention behaviors. Results from the brief survey and focus groups with CHRs about clients’ experiences, beliefs, and behaviors related to the prevention of COVID-19 were shared back with the project team and used to inform the development of the educational materials for this project [14].

### 2.2. Step 2: CHR Brief Education Development and Delivery

Three CHR Programs with long-standing relationships with the NAU Center for Health Equity Research were invited to participate in the CEAL and SHERC funded projects, including the Salt River Pima-Maricopa Indian Community, White Mountain Apache Tribe, and Colorado River Indian Tribes. A total of 15 CHRs participated.

Building from the knowledge gained through the CEAL-supported CHR focus groups, the collaborative research teams used a consensus-based decision-making methodology to review and modify vaccine education materials with CHRs, inclusive of education about the purpose of clinical research trials. Consensus-based decision-making, or a consensus panel, involves a facilitative group decision-making process [15]. In sum, to develop the materials, the project team conducted a total of two or three consensus panels per CHR Program. A total of three educational materials were created for each community: (1) COVID-19 Vaccine Information, (2) COVID-19 FAQs and Myths, and (3) COVID-19 Clinical Trials. Materials are available as supplemental files in Sears et al. [13]. Materials were provided to participants by a CHR through regularly scheduled home visits and community events. In collaboration with CHRs, pre- and post-surveys were adapted from the NIH PhenX Toolkit, a collection of COVID-19-related measurement protocols formulated in 2020, and the NIH CEAL Common Survey [11]. The surveys assessed attitudes and trust in COVID-19 vaccination and clinical trials, health messengers regarding COVID-19, and knowledge of COVID-19 vaccines, including how they work, side effects, and arranging for a vaccine. CHRs administered a pre-survey prior to providing an education session and a post-evaluation approximately 30 days after the first visit. All educational materials and study materials are available in full in this special issue [13]. Descriptive statistics were generated using SPSS software Version 10. A one-tailed paired *t*-test for significance was performed to assess mean change from pre- to post-survey. For the purposes of this paper, we focus on attitudes and planned behavior related to awareness and participation among AI community members in clinical research.

## 3. Results

### 3.1. CHR Recommendations for Culturally Relevant Communication

Of the 15 CHRs in the three CHR programs, a total of 11 CHRs participated in the CEAL focus groups in 2021. During this formative step, CHRs revealed the context necessary for understanding how to develop culturally meaningful and safe communication and educational materials about COVID-19, including vaccines and participation in clinical trial research. CHRs expressed several challenges faced by their AI clients as a result of the pandemic and described the numerous ways in which everyday life was deeply affected. Specifically, CHRs described clients being impacted at the individual, family, and community levels and by the physical and social isolation experienced during COVID-19. CHRs believed these processes led to mental health issues such as depression and anxiety as well as difficulty managing existing chronic health conditions while isolated alone at home.

During this time, CHRs also described how they directly provided support to their clients and community throughout the pandemic. Specifically, CHRs addressed the following primary fears among their AI clients: vaccine safety and potential health risks, vaccine side effects, vaccine long-term health effects, and mistrust in the federal government. CHRs raised five primary recommendations to prevent COVID-19 and improve vaccine confidence among AIs: developing culturally and linguistically relevant policies, programs, and resources to support elderly, homebound, disabled and non-English speaking communities; generate culturally and linguistically relevant mental health, isolation, grief, and loss resources; integrate COVID-19 prevention with chronic disease management, especially during periods of isolation and quarantine; improve public trust to dispel myths and correct misinformation and enhance update of public health recommendations; and invest in the professional development of the CHR workforce to promote culturally and linguistically relevant evidence-based materials and tools.

Data gathered through this first formative step was critical for the broader research teams to understand the context of our partners. SHERC research staff were invited to review and dialogue about CEAL findings which informed the development of the educational materials prior to the consensus panels. Our collective CEAL and SHERC teams also drew upon recommendations provided by CHRs involved in the CEAL focus groups regarding the ways they know clients gather, process, and act on information. CHR recommendations for designing culturally appropriate messaging and education about COVID-19 and beyond are presented in Table 1 and were essential for the SHERC team.

### 3.2. Awareness of and Attitudes toward COVID-19 Clinical Trials

As a result of CHR-led brief intervention with culturally adapted educational materials explained in detail elsewhere [13] awareness about COVID-19 clinical trials for treatment and vaccines significantly increased from baseline to post-survey among participants (Table 2). Although not statistically significant, participant knowledge of how to enroll in COVID-19-related treatment and vaccine research studies was also reported in greater frequency at follow-up compared to baseline. At 30-day follow-up, nearly 65% and 40% of participants were aware that COVID-19 vaccine and treatment trials were available, respectively.

At baseline, at least half of respondents were not willing nor likely to enroll in a COVID-19 clinical trial; only about 10% said they were willing or likely to enroll. Despite the reported low levels of interest in participating in clinical trials, participants reported recognizing the value of potential participation. At follow-up, almost two-thirds of participants reported their involvement in a COVID-19 clinical trial would make them feel like they were helping their community and helping discover treatments. In addition, participants recognized that participating in clinical trials would benefit AI/AN people outside their own communities to get COVID-19 treatment. Additionally, almost two-thirds of participants reported at follow-up that participating in a COVID-19 clinical trial for treatment would help people like them get treatment for COVID-19.

At follow-up, the top three reasons why participants would not participate in a COVID-19 clinical trial were related to not understanding what would happen to them (36%), not trusting the government (36%) or a general concern that the COVID-19 vaccine may not be safe (17%). Trusting in researchers significantly improved from pre to post, while awareness that participation in a clinical trial would not cost money significantly decreased from pre to post. Concern about a chronic condition that might prevent them from participating in research increased slightly from pre to post.

Alternatively, and although less reported as a barrier at baseline, issues related to time, transportation, importance of research and not believing vaccines are safe all de-creased from baseline to post-survey.

## 4. Discussion

Here, we describe a robust partnership between two national NIMHD-funded initiatives aimed at rapidly engaging community partnerships to address COVID-19 health inequities: CEAL and SHERC. Specifically, we describe how, through our partnership, researchers engaged the tribally employed CHR workforce of Arizona to learn best practices to engage AI community members in COVID-19 prevention and vaccine and clinical trial participation awareness. Through the Arizona CEAL project, CHRs were engaged in formative focus groups and taught researchers the community context and cultural and traditional ways of engaging Indigenous citizens in discussing COVID-19 prevention and vaccine uptake. The SHERC supplement project leveraged this local knowledge to directly collaborate with three Arizona CHR Programs to rapidly adapt and implement culturally and contextually salient educational materials to support CHRs as a trusted source to deliver brief education. Results demonstrate that although willingness to participate in a COVID-19 clinical trial for vaccines and/or treatment were mixed, there was visible progress toward trust and participation, including significant improvement in trust in researchers, awareness of existing trials and how to sign up improved through the CHR-led brief intervention. We also observed improvement in participant attitudes about the potential benefit of clinical trials, while barriers of cost of participation in research decreased.

CEAL formative focus groups also documented the multitude of ways CHRs provide accurate information to address COVID-19 and vaccine misinformation to increase the likelihood of their clients making an informed decision. CHRs described understanding that their clients and community members were afraid and lacked trust in COVID-19 information, particularly in the federal government and the safety of the vaccine. This fear and distrust may in turn dissuade participation in research generally, and especially research involving the COVID-19 vaccine or treatment. CHRs also described how, when focused on prevention, they attempted to generate confidence in COVID-19 facts. In this way, CHRs believed they reduced vaccine hesitancy and reported clients expressing a desire to get vaccinated after their concerns and questions were addressed. According to CHR participants in the CEAL focus groups, CHRs are considered trusted and influential role models within tribal communities and to their clients. CHRs believed people trust the CHR Program to provide accurate and updated information, look upon them for guidance, and follow their lead. CHRs were aware of their role model status and, therefore, dedicated time to staying informed and updated on all things related to COVID-19. In this way, they were prepared to answer client’s questions with confidence, refer them to reputable sources of information, and help them make informed decisions. Particularly for the vaccine, CHRs reported using their own experience with the vaccine to communicate with clients who may be vaccine hesitant. CHRs prioritized support of people’s autonomy and respect for their opinions and choices. Information shared by Arizona CHRs is consistent with broader United States CHW workforce involved in COVID-19 prevention and vaccine uptake efforts [12].

Our results suggest great potential of the CHR workforce and programs in addressing the social and structural determinants to research participation among underrepresented populations. Despite participants reporting less willingness to participate in clinical trials in the future, use of educational materials developed in collaboration with Indigenous and non-Indigenous researchers through consensus panels was observed to raise awareness about COVID-19-related vaccine and treatment trials available in the state among participants from pre- to post-intervention. Moreover, CHRs, through their trusted relationships and culturally relevant educational materials, were also able to improve participants knowledge of how to sign up for a clinical trial. Each of these engagement methods is in accordance with emerging research on this topic [16].

Through our research, we also learned the primary reason for not participating in a COVID-19 vaccine or treatment research among participants. These barriers included not understanding what would happen during the research, and lack of trust in government, the vaccine and/or researchers. These issues may be rooted in the social and structural issues attached to western research approaches that have historically marginalized the participation of diverse populations. Conducting culturally centered research with and for AIs represents a shift away from western research approaches and toward an Indigenous research paradigm rooted in respect and reciprocity [7].

Through culturally centered, community-based research approaches, Indigenous and non-Indigenous researchers can build stronger bridges with Native Nations to boost AI/AN representation across the United States and Canada in research studies, especially clinical trial research [7]. Emerging research suggests that when potential study participants are familiar with research team members, they are more likely to enroll in the study and complete follow-up [17]. CHRs, as trusted leaders in the community, could be a central link between Indigenous and non-Indigenous research teams and AI/AN people. Also in line with our approach to this research are documented strategies for increasing AI/AN representation in clinical trial research [7]. Strategies include engaging in tribal consultation, ensuring tribal data control and data ownership, providing timely and accurate communication, and building Indigenous research capacity through involvement of AI/AN research leadership and students in the research team [7].

At the federal level, the NIH COVID-19 Prevention Trials Network (CoVPN) [18] launched in 2020 is a significant structural policy shift within NIH aimed at correcting long-standing disparities in clinical trial participation among diverse population. CoVPN merged four existing National Institute of Allergy and Infectious Disease-funded clinical trial networks and assembled panels representative of diverse experts to review COVID-19 vaccine study protocols, including education materials and informed consent forms. CoVPN established Indigenous expert panels who identified a lack of protocol for data sovereignty, appropriate methods to engage AI/AN communities and tribal leaders, limited locations and access for AI/AN members in rural communities, and the lack of culturally tailored education tools and modalities of dissemination [18].

In sum, the strengths of this study include a diverse representation of Indigenous leadership within both the research and community partner teams, and long-standing trusted partnership between academic institutions and CHR Programs. Our limitations are important to mention as our data are not generalizable, any associations should be made with caution due to the exploratory nature of this study.

## 5. Conclusions

Full representation of diverse communities in clinical trial research is fundamental to advancing health equity. Our study demonstrates how engagement of the CHR workforce, who are known widely as the ‘pulse of the people’, may serve as agents of change. CHRs can inform efforts to effectively engage diverse and underrepresented populations in research and improve western academic institutions’ approach to ensure full representation of all people in research.

## Figures and Tables

**Table 1 ijerph-20-04391-t001:** CHR recommendations for culturally appropriate messaging and education for AI people.

1 Use plain language messaging using diagrams and images.
2 Communicate clearly about vaccine safety, efficacy, side effects, ingredients, and differences between vaccines (including both COVID-19 vaccines and the flu vaccine).
3 Include testimonies from vaccinated community members describing vaccine benefits.
4 Incorporate culturally appropriate language to describe COVID-19 prevention strategies and to talk about the vaccine to elderly clients (i.e., what is a vaccine, how it affects the immune system, and how vaccines supplement traditional medicine).
5 Use both social media (e.g., Facebook, WhatsApp) and local media (e.g., radio station, billboards) to share COVID-19 prevention and treatment information, including vaccine facts.
6 Share information from reputable institutions (e.g., CDC, local health dept.) in digestible format (illustration with minimal text).
7 Describe benefits of the COVID-19 vaccine at individual, family, community levels (e.g., ceremonies).
8 Design factsheet, pamphlets, or videos with plain information in peoples’ primary languages.
9 Collaborate with primary prevention mobile health units to reach community members in disseminating COVID-19 informational materials

**Table 2 ijerph-20-04391-t002:** Respondent awareness of and attitudes toward COVID-19 clinical trials.

	(%) of Total *n* at Baseline	(%) of Total *n* at Follow-Up
Awareness and Intention		
Are you aware of COVID-19 clinical trials that are being done? (*n* = 165) ^1^		
Yes, clinical trials for COVID-19 vaccines *	48.5	64.8
Yes, clinical trials for COVID-19 treatments *	22.4	39.4
No *	48.5	32.1
Do you know what to do to sign up for a COVID-19 clinical trial in your area? (*n* = 165; 162) ^2^		
Yes	31.5	53.1
No	68.5	46.9
How willing are you to sign up for a clinical trial for a COVID-19 treatment or vaccine? (*n* = 164; 165)		
Not at all willing	32.9	33.3
Probably not	20.1	23.0
Neutral	23.2	19.4
I’m interested	15.2	15.2
Very willing	8.5	9.1
How *likely* are you to sign up for a clinical trial for a COVID-19 treatment or vaccine? (*n* = 163; 165) ^2^		
Not at all likely	33.7	33.9
Probably not	16.0	18.8
Neutral	24.5	22.4
I’m interested	14.7	13.9
Very likely	11.0	10.9
If I were to get COVID-19, taking part in a clinical trial for a treatment would:		
Make me feel like I am helping keep other people healthy where I live (*n* = 162; 161) ^2,^*		
Strongly disagree	2.5	1.9
Disagree	5.6	7.5
Neutral	18.5	24.2
Agree	45.1	47.8
Strongly agree	28.4	18.6
Help find a treatment for COVID-19. (*n* = 164) ^2^		
Strongly disagree	1.8	1.8
Disagree	5.5	7.3
Neutral	29.3	29.3
Agree	45.1	47.0
Strongly agree	18.3	14.6
Help people like me get a treatment for COVID-19. (*n* = 144; 151) ^2^		
Strongly disagree	2.1	2.0
Disagree	5.6	7.3
Neutral	29.9	27.2
Agree	42.4	47.7
Strongly agree	20.1	15.9
Is there any reason you would NOT take part in a clinical trial for a COVID-19 vaccine or treatment? (*n* = 165) ^1,2^		
I don’t understand what will happen to me	32.7	36.4
I don’t trust the government	29.7	35.8
The COVID-19 vaccine may not be safe	18.8	17.0
I don’t trust researchers *	17.6	11.5
I have health problems that prevent me from taking part in a clinical trial	13.9	16.4
It will cost me time	12.7	12.7
I don’t have a way to get to the trial	12.1	7.9
It will cost me money *	7.9	2.4
I don’t believe clinical trials are important	4.2	1.8
Vaccines in general are bad for you	2.4	1.2
Other *	16.4	8.5

^1^ Select all that apply; ^2^ (*n* at baseline; *n* at follow-up); * demonstrates significant mean change between baseline and follow-up; one-sided *p*-value < 0.05.

## Data Availability

Data available upon reasonable request to the corresponding author. The data are not publicly available.

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
