# Peer review of "Community Health Representatives as Trusted Sources for Increasing Representation of American Indian Communities in Clinical Research"

_ijerph, 2023, doi:10.3390/ijerph20054391_

Round 1

Reviewer 1 Report

Introduction seems little long but adds lot of valuable information needed for readers to understand role of CHR. There is lot of information just on organization structure of CHR and other entities which may deter interest of readers from actual goal of your study CHR as an agent of AI/AN in COVID 19 vaccine trials. During discussion authors have provided good insight of results and roles of CHR. Some part of discussion was repetitive and can be trimmed. Conclusion supports data provided. Authors needs to provide little more information on biostatistical data analysis.

Line 23- AI/AN people- Please provide full form at first time you use abbreviation.

Line 143- “A total of 3 educational materials were created for each community: COVID-19 Vaccine Information, COVID-19 FAQs and Myths, and COVID-19 Clinical Trials. “- Please provide copy of these educational material as a supplement data for review. 

Line 165-168 -“Specifically, clients were impacted at the individual, family, and community levels – by the physical and social isolation experienced during COVID-19, which they believed had lead to mental health issues such as depression and anxiety as well as difficulty managing existing chronic health conditions while isolated alone at home. “- Is this finding from your own study if not then provide citation for it. 

Table-2- Do you have p vales for data provided? It seems like, there is not a significant change in pre and post survey for willingness and likelihood of sign up for a clinical trial for a COVID-19 treatment or vaccine when reading data from table.

Author Response

Point 1 : Introduction seems little long but adds lot of valuable information needed for readers to understand role of CHR. There is lot of information just on organization structure of CHR and other entities which may deter interest of readers from actual goal of your study CHR as an agent of AI/AN in COVID 19 vaccine trials. During discussion authors have provided good insight of results and roles of CHR. Some part of discussion was repetitive and can be trimmed. Conclusion supports data provided. Authors needs to provide little more information on biostatistical data analysis.

Response 1: Response : Thank you. We have reduced the information about CHRs in the introduction to not distract the reader and left the CHR information in the discussion. We have added indications of p values in Table 1 and this statement in the Methods :  “Descriptive statistics were generated using SPSS software. A one-tailed paired T test for significance was performed to assess change from pre to post survey. For the purposes of this paper, we focus on attitudes and planned behavior related to awareness and participation among AI community.”

Line 23- AI/AN people- Please provide full form at first time you use abbreviation.

Response 2 : Thank you. This manuscript uses terms in today’s society: Native Americans, American Indian and Alaska Native (AI/AN), Indigenous, and Native Nations. When referring the larger population, we use Indigenous and/or AI/AN. However, when referring to the study participants from Arizona, we use Native Nation, AI, or the specific community such as White Mountain Apache. We have adjusted this throughout and added the above statement to clarify our intentions and populations of focus.

Line 143- “A total of 3 educational materials were created for each community: COVID-19 Vaccine Information, COVID-19 FAQs and Myths, and COVID-19 Clinical Trials. “- Please provide copy of these educational material as a supplement data for review. 

Response 3 : Thank you. Yes, copies of these education materials are available for download from the previously published paper. We have clarified the location of these resources and cited in text : Sears, G.; Tutt, M.; Sabo, S.; Lee, N.; Teufel-Shone, N.; Baca, A.; Bennett, M.; Nashio, J.T.N.; Flores, F.; Baldwin, J. Building Trust and Awareness to Increase AZ Native Nation Participation in COVID-19 Vaccines. Int. J. Environ. Res. Public Health 2023, 20, 31. doi: 10.3390/ijerph20010031.

Line 165-168 -“Specifically, clients were impacted at the individual, family, and community levels – by the physical and social isolation experienced during COVID-19, which they believed had lead to mental health issues such as depression and anxiety as well as difficulty managing existing chronic health conditions while isolated alone at home. “- Is this finding from your own study if not then provide citation for it. 

Response 4: Yes, this information is from our own study. We have revised this section to more clearly communicate that these data are generated from CHRs involved in our study.

Based on our findings, clients were impacted at the individual, family, and community levels – by the physical and social isolation experienced during COVID-19, which they believed had led to mental health issues such as depression and anxiety as well as difficulty managing existing chronic health conditions while isolated alone at home

Table-2- Do you have p vales for data provided? It seems like, there is not a significant change in pre and post survey for willingness and likelihood of sign up for a clinical trial for a COVID-19 treatment or vaccine when reading data from table.

Response 5 : Yes, thank you, we have added indications of significance at the P values in table 2. You are correct, there were no significant change in pre and post survey for willingness and likelihood of sign up for a clinical trial for a COVID-19 treatment or vaccine.

Reviewer 2 Report

Thank you for an interesting study addressing a very important and under-addressed topic.  To make the impact higher, the following aspects should be addressed:

1. The title does not do justice to the content. Consider commencing with "Addressing Health and Cultural Inequity ..." or similar, and streamlining other aspects of the title.

2. Similarly with Keywords. Currently 'culture' is not mentioned, and the inequity is limited to vaccine.

3. In the Introduction, lines 36-40 and 41-48 in particular analyse and present key issues, but these key issues are restricted to this section. The essentials of these messages should appear in the Abstract, and in the Discussion and Conclusion.

4. The opening of the Discussion, lines 220-222, should be reframed away from describing variation in Covid-19 to addressing culturally driven Covid-19 health inequalities.  

5. Similarly, the rest of the Discussion and Conclusions should be reframed away from describing collaborative processes and programmes, to ones of reporting success in reducing health inequities and reducing mistrust of health and research systems amongst an indigenous community, through integration of programmes.

6. Please apply a spell-check, and re-read the English carefully.

Author Response

Thank you for an interesting study addressing a very important and under-addressed topic.  To make the impact higher, the following aspects should be addressed:

  1. The title does not do justice to the content. Consider commencing with "Addressing Health and Cultural Inequity ..." or similar, and streamlining other aspects of the title.

Response 1: Thank you we agree. We have adjusted the title to : Community Health Representatives as Trusted Sources for Increasing American Indian Representation in Clinical Research 

  1. Similarly with Keywords. Currently 'culture' is not mentioned, and the inequity is limited to vaccine.

Response 2: Thank you, we agree. We have adjusted the Keywords: American Indian and Alaska Native; clinical trial participation; community health representatives; cultural humility, equity

  1. In the Introduction, lines 36-40 and 41-48 in particular analyse and present key issues, but these key issues are restricted to this section. The essentials of these messages should appear in the Abstract, and in the Discussion and Conclusion.

Response 3: Thank you. These key issues have been integrated to the Abstract, and the Conclusion. Greater detail into the issues introduced in the Introduction are now found in the Discussion beginning at Line 266:

Our results suggest great potential of the CHR workforce and programs in addressing the social and structural determinants to research participation among under represented populations. Despite participants reporting less willingness to participate in clinical trials in the future, CHRs using educational materials they developed in collaboration with Indigenous and non-Indigenous researchers through consensus panels, was observed to raise awareness about COVID-19-related vaccine and treatment trials available in the state among participants from pre- to post-intervention. Moreover, CHRs, through their trusted relationships and culturally relevant educational materials, were also able to improve participants` knowledge of how to sign up for a clinical trial. Each of these engagement methods is in accordance with emerging research on this topic. (16)

Through our research, we also learned the primary reason for not participating in a COVID-19 vaccine or treatment research among participants. These issues included not understanding what would happen during the research and lack of trust in government, the vaccine and or researchers. These issues may be rooted in the social and structural issues attached to western research approaches that have historically marginalized the participation of diverse populations. Conducting culturally centered research with and for AIs represents a shift away from western research approaches and toward an Indigenous research paradigm rooted in respect and reciprocity. (7)

Through culturally centered, community-based research approaches, Indigenous and non-Indigenous researchers can build stronger bridges with Native Nations to boost AIAN representation across the United States and Canada in research studies, especially clinical trial research. (7) Emerging research suggests, that when potential study participants are familiar with research team members, they are more likely to enroll in the study and complete follow-up. (17) CHRs, as trusted leaders in the community, could be a central link between Indigenous and non-Indigenous research teams and AIAN people. Also, in line with our approach to this research, are documented strategies for increasing AIAN representation in clinical trial research. (7) Strategies include engaging in tribal consultation, ensuring tribal data control and data ownership, providing timely and accurate communication, and building Indigenous research capacity through involvement of AI/AN research leadership and students in the research team. (7)

  1. The opening of the Discussion, lines 220-222, should be reframed away from describing variation in Covid-19 to addressing culturally driven Covid-19 health inequalities.  

Response 4: Thank you. Our intention of this paper is to focus on COVID-19 health inequities which are largely structural in nature as opposed to cultural. Variations in COVID-19 were not discussed. We have added this statement in our discussion to strengthen our fous on structural barriers to participation in research among American Indian populations.

Line 238 : Results demonstrate that although willingness to participate in a COVID-19 clinical trial for vaccines and or treatment were mixed, foundational building block towards trust and participation, including significant improvement in trust in researchers, awareness of existing trials and how to sign up improved through the CHR led brief intervention. We also observed participant attitudes about the potential benefit of clinical trials improve, while barriers of cost of participation in research decrease.

  1. Similarly, the rest of the Discussion and Conclusions should be reframed away from describing collaborative processes and programmes, to ones of reporting success in reducing health inequities and reducing mistrust of health and research systems amongst an indigenous community, through integration of programmes.

Response 5: The primary outcome from the manuscript was to describe strategic ways to partner with programs such as CHRs that in-turn can reduce health inequities in the future. The collaboration with the CHRs eluted to decreased mistrust among AI respondents. The “I don’t trust researchers” variable had a significant change from baseline to post-survey; (17.6 to 11.5, respectively). We have more clearly described this result in 3.2. Awareness of and attitudes toward COVID-19 clinical trials.

  1. Please apply a spell-check, and re-read the English carefully.

Response 6: Thank you, we have copy edited the manuscript.

Reviewer 3 Report

Dear colleagues

thank you for the chance to read this paper. I find it important to highlight research with Indigenous communities. Besides a final spell check (e.g. in abstract it reads "to developed" instead of "to develop", a thorough re-read of the manuscript should fix that), I would like to suggest adding a figure relating the different parts of the research studies together. For example, you have the CEAL project and then the SHERC educational material, you have the groups and then the information sessions conducted with the developed material: I would find it helpful to see a figure where it shows me clearly how this all fits together and who was part of what.

Author Response

Reviewer 3

thank you for the chance to read this paper. I find it important to highlight research with Indigenous communities. Besides a final spell check (e.g. in abstract it reads "to developed" instead of "to develop", a thorough re-read of the manuscript should fix that), I would like to suggest adding a figure relating the different parts of the research studies together. For example, you have the CEAL project and then the SHERC educational material, you have the groups and then the information sessions conducted with the developed material: I would find it helpful to see a figure where it shows me clearly how this all fits together and who was part of what.

Response 1: Thank you. We have completed a copy edit and spell check. Addressed the abstract. Instead of a figure, we have strengthened headers within the methods section to include more clearly our steps and process.  

Line 113: 2.1. Step 1: Formative Research Through CEAL Community Health Representative (CHR) Survey and Focus Groups

Line 132: 2.2. Step 2 : CHR Brief Education Development and Delivery

Reviewer 4 Report

The authors present the results of an Arizona-based test checking if American Indians (AI) would have cultural barriers to the way Covid-19 vaccines were offered to them. Of course they have. Their native medicine is different from the Western one, but most of all there is a huge gap between an individualistic and competitive Western approach to all aspects of life and theirs, which is collective and holistic. The best strategy to overcome this gap is through understanding their culture and involving their community in the decisions and steps taken in order to vaccine them. CHRs (Community Health Representatives) were instrumental in this strategy. So the experiment can be taken as a good example to tackle other societies that are different from the individualistic Western one. By the way, the introduction also mentions AN (Alaska Natives), although as far as I have understood there were no Alaskan Natives in Arizona, where the experiment has been developed. 

I would suggest that this point about Alaskan Natives should be explained in the paper. Obviously the authors think that the points they made about American Indians (AI) should also apply to AN, but since they did not make a research with Alaskan Natives they should make it clearer.

As a whole this paper is an important contribution to the understanding of cultures that diverge from the mainstream Western ones. It also will help actions directed at them. 

Author Response

Reviewer 4

The authors present the results of an Arizona-based test checking if American Indians (AI) would have cultural barriers to the way Covid-19 vaccines were offered to them. Of course they have. Their native medicine is different from the Western one, but most of all there is a huge gap between an individualistic and competitive Western approach to all aspects of life and theirs, which is collective and holistic. The best strategy to overcome this gap is through understanding their culture and involving their community in the decisions and steps taken in order to vaccine them. CHRs (Community Health Representatives) were instrumental in this strategy. So the experiment can be taken as a good example to tackle other societies that are different from the individualistic Western one. By the way, the introduction also mentions AN (Alaska Natives), although as far as I have understood there were no Alaskan Natives in Arizona, where the experiment has been developed. 

I would suggest that this point about Alaskan Natives should be explained in the paper. Obviously the authors think that the points they made about American Indians (AI) should also apply to AN, but since they did not make a research with Alaskan Natives they should make it clearer.

Response 1: Thank you. This manuscript uses terms in today’s society: Native Americans, American Indian and Alaska Native (AI/AN), Indigenous, and Native Nations. When referring the larger population, we use Indigenous and/or AI/AN. However, when referring to the study participants from Arizona, we use Native Nation, AI, or the specific community such as White Mountain Apache. We have adjusted this throughout the manuscript and added this statement at the end of the Introduction :

Line 84 : This manuscript uses terms in today’s society: Native Americans, American Indian and Alaska Native (AI/AN), Indigenous, and Native Nations. When referring the larger population, we use Indigenous and/or AI/AN. However, when referring to the study participants from Arizona, we use Native Nation, AI, or the specific com-munity.

As a whole this paper is an important contribution to the understanding of cultures that diverge from the mainstream Western ones. It also will help actions directed at them. 

Response 2 : Thank and we appreciate your careful consideration and review.